# Physical Activity and Sedentary Behavior in Adults from Penafiel, Portugal: A Cross-Sectional Study

**DOI:** 10.3390/bs13060443

**Published:** 2023-05-24

**Authors:** Amanda Batista, Pedro Forte, Joana Ribeiro, Sandra Silva-Santos, Elmiro Silva Neto, Filipe Rodrigues, José Eduardo Teixeira, Ricardo Ferraz, Luís Branquinho

**Affiliations:** 1Department of Sports, Higher Institute of Educational Sciences of the Douro, 4560-708 Penafiel, Portugal; amandabatistagrd@yahoo.com.br (A.B.); pedromiguel.forte@iscedouro.pt (P.F.); joana.ribeiro@iscedouro.pt (J.R.); sandra.santos@iscedouro.pt (S.S.-S.); luis.branquinho@iscedouro.pt (L.B.); 2CI-ISCE/ISCE Douro, 4560-708 Penafiel, Portugal; 3Research Center in Sports Sciences, Health Sciences and Human Development, 5001-801 Covilhã, Portugal; 4Department of Sports Sciences, Instituto Politécnico de Bragança, 5300-253 Bragança, Portugal; 5Department of Sports Sciences, Federal University of Bahia, Salvador 40210-630, Bahia, Brazil; mironeto.personal@gmail.com; 6ESECS—Polytechnic of Leiria, 2411-901 Leiria, Portugal; filipe.rodrigues@ipleiria.pt (F.R.); ricardompferraz@gmail.com (R.F.); 7Life Quality Research Center, 2040-413 Leiria, Portugal; 8Polytechnic Institute of Guarda, 6300-559 Guarda, Portugal; 9Department of Sport Sciences, University of Beira Interior, 6201-001 Covilhã, Portugal

**Keywords:** epidemiology, physical inactivity, sedentary, IPAQ, exercise

## Abstract

The aim of this study was to compare the physical activity (PA) and sedentary behavior (SB) levels of young and middle-aged adults living in and around the municipality of Penafiel and to determine whether they meet PA recommendations. The researchers used the “International Physical Activity Questionnaire” (IPAQ) to measure moderate to vigorous PA and time spent on sedentary behavior (high vs. low). A prospective observational cross-sectional sample of 1105 adults aged 18–63 years, living in the municipality of Penafiel and its surroundings (45% women, 55% men), was used. The results indicated that more than half of the population was inactive (53.8%) and sedentary (54.0%). Men were more likely to be sedentary (59.2%) and inactive (55.6%) than women (inactive: 51.7%, high SB: 47.7%). Regarding daily PA and SB levels, women had higher levels of walks (3.8 ± 2.3; *p* = 0.034) and vigorous PA (2.2 ± 1.8 min; *p* = 0.005) per days/week, as well as vigorous PA per minutes/week (75.4 ± 82.1 min; *p* = 0.034). The time spent on vigorous PA per day was also higher in women (26.2 ± 22.8 min; *p* = 0.030). However, men had higher values in walking minutes per day (26.3 ± 17.1 min; *p* = 0.030), SB for weekdays (429.2 ± 141.2 min; *p* = 0.001), SB for weekends (324.7 ± 163.7 min; *p* = 0.033) and time spent on SB per minutes/week (2795.6 ± 882.0 min; *p* = 0.001). The results also showed that the older the adults, the lower the frequency and total time of vigorous PA per week. Young adults (18–28 years) had higher levels of vigorous PA (*p* = 0.005) than the other age groups (29–39; 40–50 and 51–63 years). Finally, the study found no significant correlation between individual level factors, such as number of children, marital status and monthly income, and PA or SB. Conversely, a significant and negative correlation between SB and levels of PA was found, indicating that the higher the level of PA practice, the lower the SB level. The authors suggest that promoting new PA habits and healthy lifestyles is an important future challenge for sustainability and improving the quality of life in public health.

## 1. Introduction

A healthy lifestyle associated with regular physical activity (PA) can have psychological, social and life expectancy benefits according to the World Health Organization (WHO) [1]. The practice of PA promotes an improvement in self-esteem, social acceptance and a sense of well-being [2]. In contrast, according to the American College of Sports Medicine (ACSM), sedentary lifestyle and physical inactivity are the main risk factors for the development of cardiovascular diseases, obesity, cancer and type 2 diabetes, among other diseases [3]. The evidence indicates that physical inactivity is also associated with all-cause mortality, depression, dementia, anxiety and mood swings [4], as well as a variety of other chronic diseases such as bone arthritis, osteoporosis, breast cancer and colon cancer [5].

Despite the encouragement of the importance of PA and physical exercise, the Portuguese population and world population continue to have low levels of PA. The objective, therefore, is to promote PA and physical exercise as methods of preventing cardiovascular disease and all-cause mortality [6,7]. In fact, despite the constant reinforcement of the benefits of PA by relevant organizations, physical inactivity continues to be part of the daily life of individuals and, as such, remains a significant factor of premature death [8]. It should be noted that physical inactivity and sedentary lifestyle are two different concepts. Physical inactivity refers to insufficiencies of PA and not meeting specific recommendations. Conversely, sedentary behavior is defined as any waking behavior characterized by the expenditure of 1.5 metabolic equivalent tasks (METs) or less of energy while in a sitting, reclining or lying posture [9].

Physical inactivity and sedentary lifestyle are strongly related to the incidence and severity of a vast number of chronic diseases [10], contributing to the decline in the general state of health. Considering the high prevalence rate of inactive and sedentary individuals in developed countries [11,12], several diseases are becoming increasingly prevalent in society. The advancement of new technologies is a preponderant and influential factor of low PA. The main sedentary behaviors are: watching television, playing video games and sitting too long [12]. Additionally, the low energy expenditure resulting from sedentary tasks associated with mental demands, especially those arising from work, leads to increased food consumption that results in a positive energy balance [13,14].

From the WHO to the European Union (EU), from the Portuguese Government to local authorities, there are several references, plans and recommendations to promote an active lifestyle, aiming the improve health and quality of life. As a form of intervention, the WHO recommends at least 150 min of moderate to vigorous PA per week, equivalent to 30 min per day. However, such recommendations are not fulfilled [12]. In Portugal, the municipality of Penafiel established a series of partnerships with local sports associations, with the aim of promoting and encouraging the practice of sports and the generalization of physical activities in the municipality. Other measures implemented by the municipality are the programs for the use of sports spaces, the programming of physical and sports activities aimed at the general population, as well as the carrying out of training and qualification actions for sports agents.

According to the National Institute of Statistics (INE), the municipality of Penafiel, located in the north of Portugal, in the district of Porto, has a population of around 72,000 inhabitants (338.4 inhabitants per km^2^) and is made up of 28 parishes. Understanding the levels of PA and SB for this highly populated area is crucial to cross-check the overall data of European epidemiological reports (Eurobarometer, 2022). In this sense, the objective of this study was to know and compare the level of PA and sedentary behavior of young and middle-aged adults living in the municipality of Penafiel and its surroundings. Furthermore, it aimed to verify if the PA recommendations are performed.

## 2. Materials and Methods

### 2.1. Sample

A prospective observational cross-sectional analysis was conducted using a sample of 1105 adults, with a mean age of 28.7 ± 11.5 years, living in the municipality of Penafiel and its surroundings. Of the sample, 45% was formed by women and 55% by men, aged between 18 and 63 years. For inclusion, those who met the following inclusion criteria were considered: (i) aged 18 years old or older; (ii) provided informed consent to participate; and (iii) were physically active. We verified that the marital status of 64.7% of the sample was single, 31.9% married, 2.4% divorced and 1% widowed. In addition, 67.2% had no children, 15% had one child, 13.8% had two children and 3.8% had three or more children. Finally, the monthly income of 57.6% of the sample was less than EUR 700, 29.5% had an income between EUR 700 and 1200 and only 9.6% and 3.3% presented an income of EUR 1201 to 1700 and above EUR 1700, respectively. Thus, most of the participants in this study were single (64.7%), without children (67.2%) and with monthly income of less than EUR 700 (57.6%) [15].

In this study, we determined the required sample size using G*Power 3.4 (Institut für Experimentelle Psychologie, Düsseldorf, Germany). The parameters used were an anticipated effect size of f = 0.03, α = 0.05 and statistical power of 0.95, which resulted in a minimum sample size of 253 participants per group. This sample size was met in the study.

### 2.2. Procedures

The data collection was performed through an online PA questionnaire: “International Physical Activity Questionnaire” (IPAQ)—short version, using the Google Forms online platform. The questions were preceded by a brief explanation of the procedures in order to clarify all possible doubts of the participants. The use of a small and simple questionnaire in an online format with all the necessary information and clarifications provides greater accessibility and consequently the possibility of a larger sample size. A previous study reported a high reliability for the IPAQ with a reliability test–retest (ICC) of 0.90 and concurrent validity (PC) of 0.917 to assess physical and sedentary behavior in European adults [16].

The IPAQ short version is composed of seven questions that evaluate physical activity across various areas such as leisure time, work-related activity and transport-related activity. The first two questions inquire about the number of days and duration of time spent engaging in moderate-intensity activities such as brisk walking or cycling at a relaxed pace, while the subsequent two questions assess the number of days and duration of time spent engaging in vigorous-intensity activities such as running, cycling at a fast pace or heavy lifting. Question five assesses the amount of time spent sitting during a typical weekday and question six assesses the amount of time spent sitting during a typical weekend day. The final question asks about the frequency of walking or cycling for transport-related purposes. Responses to these questions are used to determine the total amount of physical activity per week and the amount of time spent on moderate- and vigorous-intensity activities separately. The IPAQ is a reliable and valid tool for assessing physical activity levels among various populations and settings. This tool has become a popular choice for researchers, health professionals and educators alike. Its purpose is to gauge the volume and intensity of physical activity performed in various settings, including home, work, leisure and other activities within the general population [1,17]. The total sample was divided into four age groups: (1) 18–28 years, (2) 20–39 years, (3) 40–50 years and (4) 51–63 years. The questionnaire utilized in this study comprised five sections, namely personal data, walks, moderate activities, vigorous activities and no activities. Participants were grouped into four categories: (1) physically active, defined as engaging in vigorous PA for more than 75 min per week or moderate PA for more than 150 min per week; (2) physically inactive, defined as engaging in vigorous PA for 75 min or less per week or moderate PA for 150 min or less per week; (3) high sedentary behavior, defined as spending 8 or more hours per day in a sitting, reclining or lying posture; and (4) low sedentary behavior, defined as spending less than 8 h per day in a sitting, reclining or lying posture [16].

### 2.3. Statistical Analysis

Statistical analysis was performed using the Statistical Package for Social Sciences (SPSS 27.0, IBM Corp, Armonk, NY, USA). The significance level was set at 5%. Descriptive statistics were performed using the mean, standard deviation and median values. A Mann–Whitney test was applied in three different comparisons: number of days in the last week in which participants walked or carried out moderate PA and vigorous PA for at least 10 min; total time (in minutes) in which walks, moderate PA and vigorous PA were performed; time spent in a sitting, reclining or lying posture during a weekday and during a weekend day. Furthermore, we compared the same variables by four age groups through a Kruskal–Wallis test. Finally, the Spearman rank correlation (r) test was performed to perceive the existence and degree of correlation between the variables of SB, PA and personal data (number of children, marital status and monthly income). The correlation magnitude was classified as: trivial if r ≤ 0.1, small if r = 0.1–0.3, moderate if r = 0.3–0.5, large if r = 0.5–0.7, very large if r = 0.7–0.9 and almost perfect if r ≥ 0.9 [18].

## 3. Results

Table 1 presents the mean differences between men and women in levels of physical activity (PA) and sedentary behavior (SB), with statistically significant differences found for walks (*p* = 0.034 to 0.042), vigorous PA (*p* = 0.005 to 0.030) and time spent on SB (*p* = 0.001 to 0.033). Women reported higher frequency of walks (3.8 ± 2.3; *p* = 0.034) and vigorous PA (2.2 ± 1.8 min; *p* = 0.005) per days/week compared to men. While women spent more time on vigorous PA per day (26.2 ± 22.8 min; *p* = 0.030), men reported higher values for walking minutes per day (26.3 ± 17.1 min; *p* = 0.042), SB on weekdays (429.2 ± 141.2 min; *p* = 0.001) and SB on weekends (324.7 ± 163.7 min; *p* = 0.033). Additionally, women showed better levels of vigorous PA per minutes/week (75.4 ± 82.1 min; *p* = 0.034), whereas men spent more time on SB per minutes/week (2795.6 ± 882.0 min; *p* = 0.001).

Table 2 presents the frequency of PA and SB over the monitored period. More than half of the overall population was inactive (53.8%) and sedentary (54.0%). Men sampled tended to be more sedentary (59.2%) and inactive (55.6%) than women (inactive: 51.7%, SB: 47.7%).

Table 3 presents the levels of PA practice and SB by age groups: (1) 18–28 years; (2) 20–39 years; (3) 40–50 years and (4) 51–63 years. We observed significant differences only between group 1 (18–28 years) and the other groups in levels of vigorous PA: number of days/week (*p* = 0.001), number of minutes/day (*p* = 0.007) and number of minutes/week (*p* = 0.005). Group 1 presented higher levels of vigorous PA per days/week (2.2 ± 1.9 days; *p* = 0.001 to 0.021), per minutes/day (26.9 ± 21.2 min; *p* = 0.010 to 0.045) and per minutes/week (76.4 ± 84.3; *p* = 0.10 to 0.032) than groups 2, 3 and 4.

According to Spearman rank correlation analysis, significant correlations were observed between PA or SB and the individual level factors: number of children (*p* < 0.05 to *p* < 0.001), marital status (*p* < 0.05 to *p* < 0.001) and monthly income (*p* < 0.05). Thus, these variables do not seem to be related to levels of PA and SB. On the other hand, significant and negative correlations between SB (time spent in a sitting, reclining or lying posture) and levels of PA per minutes/week were found for walks (r = −0.65; *p* = 0.001), moderate PA (r = −0.409; *p* = 0.001) and vigorous PA (r = −0.508; *p* = 0.001).

## 4. Discussion

The purpose of the current study was to investigate and compare the levels of physical activity (PA) and sedentary behavior (SB) of young and middle-aged adults residing in the municipality of Penafiel and surrounding areas, and to determine if they meet the recommended PA guidelines. The findings indicated that over half of the population did not meet the minimum recommended levels of PA (53.8%) and had high levels of sedentary behavior (54.0%), which confirms the high prevalence of physical inactivity and sedentary behavior reported in previous epidemiological studies [18,19]. In this sense, the study’s subjects of research are at an increased risk for several chronic conditions, such as the development of several diseases (e.g., obesity, cancer, type 2 diabetes, cardiovascular diseases and depression), as well as lower cognitive development and mortality [3,19]. Evidence supports that PA is beneficial for the prevention of several types of cancer such as breast, colon, endometrium, kidney, bladder, esophagus and stomach. Furthermore, minimizing time spent in SB may also contribute to decreasing the risk of metabolic diseases endometrial, colon and lung cancer [20,21,22]. Thus, the results found in this study serve as an alert to promoters of PA and health in Penafiel, as they demonstrate a great risk to public health in the region. PA and physical exercise (PE) seem to be the best tools to combat this problem and government action is essential.

The results verified that the men sampled tended to be more sedentary and inactive than women. Juren et al. [22] evaluated the total daily sitting time for male and female undergraduate students and compared their daily sitting time between weekdays and weekends. The authors observed similar results (*p* = 0.169) for female undergraduates ($\bar{x}$ = 9.6 h/day) and male undergraduates ($\bar{x}$ = 9.5 h/day). Prince et al. [21] investigated the Canadian SB time statistics in 2019 and verified that Canadian adults aged 18–79 spent an average of 573 min/day on SB (men: 574 min/day and women: 572 min/day). The SB was analyzed by age group and the results showed that among youth, females were more sedentary than males in the earliest (2007–2009; 9.4 vs. 8.9 h/day, *p* = 0.009) and most recent cycles (2016–2017; 9.2 vs. 8.5 h/day, *p* = 0.003). 

Among adults aged 35–49 years, women had significantly higher levels of SB than men from 2007–2009 (9.7 vs. 9.4 h/day, *p* = 0.037) and 2009–2011 (9.9 vs. 9.6 h/day, *p* = 0.030), but not in subsequent cycles. Furthermore, no significant gender differences were found in adults aged 18–34 years, 50–64 years or ≥65 years. According to this study, young men (12–17 and 18–34 years) spend significantly more leisure time playing video games than young women. Another factor analyzed is related to the use of a car as a means of transport. The authors indicate that women drive less and are more likely to be passengers in vehicles or use public transit. Another study compared the sitting time between males and females [22]. The authors stated that men and women with same occupation may perform different work tasks, which will result in different patterns of sitting time and PA at work. Thus, this study concluded that men spend on average 72% and women 67% of their worktime sitting. About 50% of the men and 34% of the women spent more than 75% of their time at work sitting, predominantly in uninterrupted bouts longer than 30 min, which is also consistent with findings by Hadgraft et al. [23].

A systematic review suggests that SB and time spent on various sedentary activities differ by gender in adults (18–64 years). Women are negatively associated with total sitting, television and screen entertainment and passive travel when compared to men [24]. On the other hand, Edwards and Sackett [25] conducted a study to review psychosocial influences on women’s participation in PA because they believed that women have a lower level of PA compared to men. Self-efficacy, social support and motivation are empirically substantiated factors that explain the lower rates of participation in PA of women than men. According to several authors [19,26,27], prolonged durations of daily sitting time (more than 6 h daily) have a negative impact on health-related quality of life because they are associated with higher rates of chronic diseases and premature death, especially among working adults. Furthermore, the lack of physical activity is the fourth leading cause of death in the world [28].

In the PA analysis, the outcomes were divided by age group: (1) 18–28 years; (2) 20–39 years; (3) 40–50 years and (4) 51–63 years. We verified that group 1 presented higher levels of vigorous PA than groups 2, 3 and 4. Thus, the results showed that the greater the age, the lower the frequency and total time of vigorous PA per week. Unlike our results, a study of Finnish adults which was carried out from 1972–2002 showed that PA tended to increase with age [29]. The young adults (18–29 years) were the only group to present adequate mean values of vigorous PA per week (76.4 ± 84.3 min/week), according to specific WHO recommendations (vigorous PA > 75 min/week). However, no group obtained, on average, adequate moderate PA within the recommended values (moderate PA > 150 min/week). The best result of moderate PA was achieved by group 3 (40–50 years), with 93.6 ± 107.6 min/week. On the other hand, the worst results of moderate and vigorous PA were observed in group 4 (51–63 years) with 81.7 ± 88.2 min/week and 60.3 ± 89.0 min/week, respectively. Dagmar et al. [27] performed a study to describe the changes in PA, SB and BMI of the inhabitants in the Liberec region (Czech Republic) from the years 2002–2009 and compared the results of women and men. Regardless of the year of monitoring and gender, there were no significant differences in PA among age groups (age brackets 25–35, 36–45 and 46–60 years). Furthermore, men show more PA in total than women, which can be explained mainly by the differences in vigorous PA [30,31,32].

In the last analysis, no significant correlations were observed between PA or SB and the individual level factors: number of children, marital status and monthly income. Therefore, variables do not seem to be related to levels of PA and SB. On the other hand, O’Donoghue et al. [24] verified a significant correlation between SB and individual level factors (body mass index, socio-economic status and mood). The authors observed that the married or cohabiting participants had a trend towards increased amounts of leisure screen time while having children resulted in less total sitting time. We also found significant and negative correlations between SB and levels of PA. Therefore, the results suggest that the higher the PA level, the lower the SB level. Furthermore, there were significant and positive correlations between PA practice and environmental characteristics (including proximity of green space, neighborhood walkability, safety and weather), thus emphasizing the important role of the environment in the practice of PA. In this sense, efforts are needed to promote public health through spatial planning and policy interventions. Thus, a well-designed built environment that promotes PA would help reduce levels of physical inactivity and promote public health in a low-cost and effective way [33]. 

According to the Eurobarometer of Sport and Physical Activity (published by the European Commission in 2018 and 2022), there was an increase in the prevalence of physical inactivity in Portugal [30]. The Eurobarometer of 2022 showed that Portugal is the country of the European Union with the most respondents that say they never exercise or play sport (73%). Furthermore, the study verified a higher proportion of inactive people in Portugal compared to the Eurobarometer of 2018 [31,32]. Specifically in Penafiel, the scenario does not seem to be different, considering the results presented in this study. Therefore, in addition to plans and programs of action, it is essential to perform a regular and repeated monitoring of PA at national and regional levels. The reversing of this trend is a national challenge that requires a multisectoral strategy, with government support and support for the implementation of concrete actions, especially in a Portuguese context.

There are several limitations to consider in this study. Firstly, the data collected are based on self-reported responses, which may be subject to recall bias and may not accurately reflect participants’ actual levels of physical activity and sedentary behavior. Additionally, the cross-sectional design of the study only allows for the observation of associations between variables and does not establish causality. Socio-economic variables were not analyzed, which may have an impact on physical activity and sedentary behavior and may limit the generalizability of the findings to other populations. Lastly, another limitation of this study is that body mass index was not assessed, which could provide a more comprehensive understanding of the relationship between physical activity levels and risk factors for chronic diseases. Future research should consider incorporating BMI measurements to further explore the impact of physical activity on health outcomes.

## 5. Conclusions

The study revealed that a significant proportion of young and middle-aged adults residing in the Penafiel municipality fall short of meeting the recommended minimum levels of physical activity (53.8%) and have a high prevalence of sedentary behavior (54.0%). Thus, despite numerous national and international recommendations, society is still not active enough, probably as consequence of the current pace of people’s lives and all the transformations brought about by technology. Urgent action is required from the city council and relevant organizations to create and promote public health interventions aimed at increasing physical activity and promoting healthy lifestyles. It is crucial to prevent the negative health outcomes associated with physical inactivity and sedentary behavior. Encouraging the population to adopt healthier habits and behaviors is vital for ensuring sustainability and enhancing overall quality of life. Although this study did not analyze SES variables, it is important to acknowledge the potential influence of SES on PA and SB behaviors, and further investigation is needed in this area. 

## Figures and Tables

**Table 1 behavsci-13-00443-t001:** Mean differences among sexes in physical activity practice and sedentary behavior levels.

Variables		Men (*n* = 608)			Women (*n* = 497)			Total			Proof Value
		Mean	SD	Median	Mean	SD	Median	Mean	SD	Median	
Number of days/week	Walks	3.5	2.1	3.0	3.8	2.3	4.0	3.6	2.2	4.0	0.034 *
	Moderate PA	2.6	2.1	3.0	2.7	2.1	2.0	2.7	2.1	2.0	0.615
	Vigorous PA	2.0	2.1	2.0	2.2	1.8	2.00	2.1	2.0	2.0	0.005 *
Number of minutes/day	Walks	26.3	17.1	20.0	24.6	17.1	25.0	25.6	17.1	20.0	0.042 *
	Moderate PA	25.7	20.1	25.0	27.9	22.1	30.0	26.7	21.5	25.0	0.248
	Vigorous PA	24.2	25.3	20.0	26.2	22.8	20.0	25.1	24.2	20.0	0.030 *
	SB on weekday	429.2	141.2	480.0	395.9	151.9	480.0	414.2	147.0	480.0	0.001 *
	SB on weekend	324.7	163.7	333.0	303.8	155.2	300.0	315.3	160.2	300.0	0.033 *
Number minutes/week (number days * minutes/day)	Walks	97.0	79.6	90.0	102.8	88.9	80.0	99.6	83.9	80.0	0.495
	Moderate PA	86.8	88.9	60.0	97.5	104.5	60.0	91.6	96.3	60.0	0.287
	Vigorous PA	70.6	89.6	60.0	75.4	82.1	45.0	72.8	86.3	50.0	0.034 *
	SB during week	2795.6	882.0	3120.0	2587.0	943.1	3200.0	2701.8	915.4	3180.0	0.001

Notes: SD: standard deviation; Min: minimum value; Max: maximum value; PA: physical activity; SB: sedentary behavior; SB on weekday: time spent in a sitting, reclining or lying posture on a weekday; SB on weekend: time spent in a sitting, reclining or lying posture on a weekend day; * *p* ≤ 0.05: significant differences.

**Table 2 behavsci-13-00443-t002:** Physical Activity and Sedentary Behavior Frequency.

Group	Moderate and/or Physical Activity (*n* (%))		Sedentary Behavior (*n* (%))	
	Active ^1^	Inactive ^2^	High ^3^	Low ^4^
Women	240 (48.3%)	257 (51.7%)	237 (47.7%)	260 (52.3%)
Men	270 (44.4%)	338 (55.6%)	360 (59.2%)	248 (40.8%)
Total	510 (46.2%)	595 (53.8%)	597 (54.0%)	508 (46.0%)

Notes: ^1^ Active: vigorous PA > 75 min/week or moderate PA > 150 min/week; ^2^ Inactive: vigorous PA ≤ 75 min/week or moderate PA ≤ 150 min/week. Sedentary behavior: ^3^ high sedentary behavior: number of hours in a sitting, reclining or lying posture ≥ 8 h/day; ^4^ low sedentary behavior: number of hours in a sitting, reclining or lying posture < 8 h/day.

**Table 3 behavsci-13-00443-t003:** Levels of physical activity practice and sedentary behavior by age groups.

Variables		18–28 Years (*n* = 656)			29–39 Years (*n* = 230)			40–50 Years (*n* = 158)			51–63 Years (*n* = 61)			Proof Value
		Mean	SD	Median	Mean	SD	Median	Mean	SD	Median	Mean	SD	Median	
Number of days/week	Walks	3.7	2.1	4.0	3.4	2.3	3.0	3.5	2.2	3.0	3.3	2.4	3.0	0.146
	Moderate PA	2.7	2.0	3.0	2.6	2.2	2.0	2.7	2.2	2.0	2.7	2.3	2.0	0.775
	Vigorous PA	2.2	1.9	2.0	2.0	2.1	1.0	1.9	2.1	1.0	1.4	1.9	1.0	0.001 *
Number of minutes/day	Walks	25.2	16.6	20.0	26.0	18.7	25.0	26.6	17.0	25.0	25.1	16.9	25.0	0.854
	Moderate PA	26.9	21.2	25.0	27.6	22.9	30.0	26.1	21.8	30.0	22.5	18.4	20.0	0.449
	Vigorous PA	26.9	23.9	20.0	22.7	24.2	20.0	21.5	21.9	20.0	24.0	31.1	15.0	0.007 *
	SB on weekday	410.0	144.5	480.0	413.4	153.1	480.0	418.4	156.3	480.0	452.8	119.7	480.0	0.084
	SB on weekend	313.1	166.7	300.0	317.7	148.7	300.0	313.4	151.4	300.0	335.3	155.6	360.0	0.681
Number minutes/week (number days * minutes/day)	Walks	100.2	80.4	90.0	99.9	92.3	75.0	99.4	83.2	80.0	91.8	91.4	60.0	0.387
	Moderate PA	89.9	92.9	60.0	97.9	99.8	80.0	93.6	107.6	60.0	81.7	88.2	60.0	0.839
	Vigorous PA	76.4	84.3	60.0	71.1	94.1	45.0	65.0	81.2	30.0	60.3	89.0	15.0	0.005 *
	SB during week	2675.9	918.9	3180.0	2702.1	926.8	3180.0	2718.8	935.2	3180.0	2934.4	756.6	3240.0	0.567

Legend: SD: standard deviation; Min: minimum value; Max: maximum value; PA: physical activity; SB: sedentary behavior; SB on weekday: time spent in a sitting, reclining or lying posture on a weekday; SB on weekend: time spent in a sitting, reclining or lying posture on a weekend day; * *p* ≤ 0.05: significant differences.

## Data Availability

Data are available upon request from the corresponding author.

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
