# Peer review of "Physical Activity and Sedentary Behavior in Adults from Penafiel, Portugal: A Cross-Sectional Study"

_behavsci, 2023, doi:10.3390/bs13060443_

Round 1

Reviewer 1 Report

General comments

This study mainly aimed to examine the level of PA and sedentary behaviour of young and middle-age adults living in the municipality of Penafiel and sur-roundings.

The theme of the article is relevant and the topic is very important. Whilst the article is interesting, there are a few flaws and several points are worth addressing.

Title

In my opinion title should not beginning with Analysis of the level .... Analysis is the process authors make with results obtained from data using different statistical methods. In addition, title doesn't suggest comparisons between groups of the adults separated based on age, what is stated in the aim of the work. I suggest for examiple: The differences/disparities in PA and SB between young and middle-age .... or The level of PA and SB in young comparing to middle-age adults ........  Or maybe, The Effect of the Age on PA and SB in adults living in the municipality of Penafiel, Portugal: A cross-sectional research. I suggest also A cross sectional study (instead of research).

Aim of the work

Aim is well-written, but there are mistakes in aim in main text and Discussion paragraphs (it's a pity there aren't lines): compare the level of PA and sedentary behaviour of young and middle adults living..... Should be middle-age adults.

Sample

Sample size should be calculated. Why authors examined 1105 persons. The number was adequate or maybe it was excessive number of participants? It should be supplemented.

Besides, who was the participants. What was including or/and excluding criteria (besides age: 18 years of age). What was the procedure of recruiting the participants? We know only the socio-demographic variables obtained from recruited participants.

Procedures

We don't know the way the data were obtained? Well-trained staff inteviewed the participants or online data from questionnaires were collected? These both ways of acquiring data have pros and cons. Authors should presented the method and explained their choice. 

IPAQ should be described more detailed. Reliability and accuracy should be included.

Statistical analysis

In my opinion there is inconsistency in statistical methods used by Authors. I assume there were no normal distribution of the data, because Authors used non-parametrical methods to compare groups of participants. So, why Pearson correlations were calculated instead of Spearman rank-correlations? It should be explained (with arguments why) or recalculated. 

Analysis and Discussion

The parts related to analysis of the relationships should be updated after recalculation. Limitations and strenghts should be included.

Conclusions

Although socio-economic variables were not analyzed, the conclusions should indicate the need to study the differentiation of PA and SB in subgroups with different SES status.

Author Response

Dear authors,

We appreciate your feedback. Revisions were made and tracked using the track-change option in MS Word.

Please, see attachment.

Thanks in advance.

With best regards.

José Eduardo Teixeira.

Reviewer 2 Report

Dear Authors,
You have addressed a very important research issue. The article is interesting and the problem is topical. As you have noted, despite numerous recommendations - including those of the WHO - society is still not active enough.
The conclusions in the article also confirm this. It is worth looking for reasons for this state of affairs - simply stating that "the young and middle-aged adults living in the Penafiel municipality does not comply with the recommended minimum levels of PA (53.8%) and have a high time spent in SB (54.0%)" seems insufficient. It is worth indicating 'something' more in the conclusions to enhance its value.

- In my opinion, for the data analysed it's useful to present the results using the median - mean values can distort the interpretation (see high SD, Min-Max values)

- The authors write "Therefore, we verified that the higher the PA practice, the smaller the SB level". - this seems obvious. The more active we are, the less time we spend sedentary 

- page 5 - please remove the blank space (there are only 3 lines)

- Table 2, in my opinion, does not show the frequency versus percentage of active and inactive subjects. The results presented do not give a complete picture - what percentage of the total study population was active (high/moderate) versus sedentary. Now the percentages add up in these areas.

- Did the surveys have no limitations???? I assume they had - please add in the article.

- The authors in the discussion write "In this sense, the study's subject research is at an increased risk for several chronic conditions, as the development of several diseases (e.g., obesity, cancer, type 2 diabetes, cardiovascular diseases and depression), as well as lower cognitive development and mortality [3,20]." In future work, it is also useful to assess BMI - when combined with physical activity levels, it may give a better picture of risk factors.

Author Response

(The authors gave the same response as above.)

Round 2

Reviewer 1 Report

Almost done. But I suggest to supplement the  Table 1 with Median values. It is because Authors conducted non-parametrical analyses (Mann-Whitney, Kurskal-Wallis, Spearman). 

Author Response

Dear Reviewer,

We appreciate your feedback. Thank you very much.

As requested we have added the median values in table 3 and 4, as well as in statistical analysis section. 

With best regards,

José Eduardo Teixeira.